# How Can China and the Belt and Road Initiative Countries Work Together Responding to Climate Change: A Perspective on Carbon Emissions and Economic Spillover Effects

**DOI:** 10.3390/ijerph19159553

**Published:** 2022-08-03

**Authors:** Yanmei Li, Xiushan Bai

**Affiliations:** School of Economics and Management, Beijing University of Technology, Beijing 100124, China; baixs@emails.bjut.edu.cn

**Keywords:** climate change, green development of the belt and road, carbon emissions spillover effects, economic spillover effects, multi-regional input-output analysis

## Abstract

China attaches great importance to international cooperation on climate change, especially working with the Belt and Road Initiative relevant partners. From a perspective on carbon emissions and economic spillover effects, this study explores how China and the Belt and Road Initiative countries can work together to cope with climate change. It applies a three-region spillover effects model, based on the multi-regional input-output table and satellite extensions data of the Eora database, to examine mutual carbon emissions and economic spillover effects between China and the Belt and Road Initiative countries. The results show that: (1) Mutual carbon emissions spillover effects exist between China and the BRI countries, which is an important premise for them to cooperate responding to climate change. (2) There are great differences in carbon emission spillover effects among different sectors. Thus, optimizing the trade structure can bring benefits to carbon reduction for both sides. (3) The sectoral order of carbon emissions spillover effects, and economic spillover effects, is not consistent. In order to achieve an economic and environmental win-win, it is necessary to increase bilateral trade in those sectors with large economic spillover effects, and reduce other sectors with large carbon emissions spillover effects.

## 1. Introduction

With economic activities strengthening, the resulting effects on climate change have emerged as a threat to the long-term, stable, and sustainable development of social and eco-systems. On 4 April 2022, the United Nations Intergovernmental Panel on Climate Change (IPCC) issued the third part of the Sixth Assessment Report, Climate Change 2022: Mitigation of Climate Change, the Working Group III contribution. In the past decade, global greenhouse gas emissions have reached unprecedented heights, but the growth rate is declining [1]. Balancing the relationship between economic development and climate change has become an urgent focus of global conversations. As an important factor affecting climate change, the massive increase of carbon dioxide emissions is a severe and immediate global problem. Global carbon emissions have continuously grown since 2013, reaching a record high of 34.36 billion tons in 2019 [2].

In response to the mounting climate crisis, more than 100 countries around the world have established carbon neutrality goals [3]. Among them, China is one of the most active countries to promote carbon neutrality. On 22 September 2020, China first proposed a vision of achieving the carbon peak by 2030, and carbon neutrality by 2060, at the 75th General Debate of the United Nations General Assembly [4,5]. On 27 October 2021, China released a white paper, Responding to Climate Change: China’s Policies and Actions. The white paper emphasizes that China attaches great importance to jointly building the Green Silk Road with other countries, in order to address climate change. [6]. Since the Belt and Road Initiative (BRI) was proposed in 2013, trade links between China and participating countries have become closer and closer. The total import and export of products between China and the BRI countries increased from US$1.04 trillion to US$1.34 trillion between 2013 and 2019, with a growth rate of 28.85% [7].

On 12 December 2015, nearly 200 parties to the United Nations Framework Convention on Climate Change reached the Paris Agreement at the Paris Climate Change Conference. In this context, the issue of carbon emissions has received widespread attention from various countries. Trade can promote economic growth and generate embodied carbon emissions [8,9,10,11,12,13]. This paper aims to address the issue of the cooperation between China and BRI countries to tackle climate change, from the perspective of trade-related economic and carbon emission spillover effects. We address two questions: First, what kind of spillover effects will trade between China and BRI countries have on their carbon emissions and economies? Second, from the perspective of weighing carbon emissions and economic spillover effects, how can trade cooperation between China and BRI countries work together to address climate change? The novelty of this paper mainly lies in two aspects. First, from the perspective of trade-related spillover effects, this paper analyzes the correlation of economy and carbon emissions between China and BRI countries. Second, on the basis of comparing the differences of economic and carbon emission spillover effects in different sectors, this paper explores the policies for China and BRI countries to cooperate when dealing with climate change.

The rest of the paper is organized as follows. Section 2 briefly reviews the existing related literature. Section 3 presents methodology and data sources. Section 4 analyzes and discusses the results. Section 5 outlines main conclusions, puts forward some policy suggestions, and points out the direction for future research.

## 2. Literature Review

### 2.1. Response to Climate Change under the Belt and Road Initiative

Since the Belt and Road Initiative (BRI) was launched in 2013, economic and trade cooperation among participating countries has continued to strengthen [14,15]. Noting that “win-win” is the core goal of the BRI [16], many studies have focused on whether a mutually beneficial situation of economy and environment can be achieved by participating countries. In particular, concerns over the BRI’s potential impacts on environment as well as global climate change, have gathered force [17]. Some studies raised concerns about possible carbon emissions impacts of the BRI [18,19]. They believe that the Belt and Road Initiative may worsen the environment of the countries along the route, and even lead to “energy plunder” [18,19].

Other studies found that the BRI will promote carbon emission reduction and environmental improvement of participating countries, through trade and investment [20,21]. In particular, the countries participating in the Belt and Road Initiative have expressed positive views on this issue. Siti Nurhasanah (2019), an Indonesian scholar, believes that the project has benefited all participants and helped poor areas [22]. Lao scholar Visansack Khamphengvong (2022) said that the Belt and Road Initiative plays a crucial role in mobilizing economic and social development, and improving national and international connectivity [23]. Other scholars have also argued for the positive effects of the Belt and Road Initiative in terms of economic development [24], ecological protection [25], and food security [26]. In the context of the growing trade between BRI countries, carbon emissions embodied in trade have become the focus of discussion. The volume of carbon emissions embodied in trade has been measured at the national or regional level in BRI countries [27]. The factors which affected carbon emissions embodied in trade changes have been analyzed by using structural decomposition analysis method [28,29,30].

Carbon emission reduction under the BRI is of great significance to China and other participating countries in addressing climate change [31]. If China and the BRI carry out green cooperation, it will not only help the two sides to jointly cope with climate change [32], but also benefit the whole world in many ways [33]. Greening infrastructure and energy investments in the BRI will have an important impact on carbon emission reduction globally [34]. Promoting transition to a low-carbon energy system is an important measure for China and BRI countries to deal with climate change together [35]. In short, promoting international cooperation between China and BRI countries through deepening green trade and investment is of great significance to tackling global climate change.

### 2.2. Economic and Carbon Emissions Spillover Effects

Spillover and feedback effects are important manifestations of inter-regional interactions. Economic spillover effects are the unidirectional influence of the economic development of region A on the economic development of other regions. Feedback effects refer to the influence of other regions on the development of region A [36]. Carbon emission spillover and feedback effects are the products of economic interaction between different countries, or regions. Spillover and feedback effects models were first proposed by Miller [37,38] and gradually developed by scholars such as Round [39,40,41] and Sonis [42,43,44,45], who further clarified the economic meaning of intra-regional multiplier effect, inter-regional spillover effect, and inter-regional feedback effect. He pointed out their interrelationships and the consistency of multiplicative and additive decompositions. Furthermore, he proposed a unified approach to measure various types of effects using final demand as the starting point. Finally, Zhang [46,47] extended spillover and feedback effects to the value-added perspective, and discussed inter-regional interactions on the supply and demand sides, respectively.

Initially, spillover and feedback effect models were mainly used to analyze the interlinkage effects of economies between different areas. Later, with the growing prominence of environmental issues, spillover and feedback effects models were expanded to the environmental domain [48]. Some scholars have discussed the interactions of environmental factors such as carbon, water, and land among different regions in a country [49,50], or different countries [51]. It is found that the spillover effects are much larger than the feedback effects [52,53,54]. Interregional trade contributes to economic output, as well as causing significant carbon emissions. Some studies have begun to discuss the relationship between economic spillover effects and carbon emissions spillover effects [55]. A few studies have focused on the economic or carbon emissions spillover and feedback effects in BRI countries. Wang et al. [56] found an overall upward trend in the contribution of China’s economic development to the economic growth of BRI countries, with regional heterogeneity among countries. Wang et al. [57] presented that the mutual carbon spillover effects differed among seven BRI regions.

Although existing research can provide reference for the formulation of international cooperation policies to deal with climate change under the BRI, there are still two aspects of limitation. On the one hand, there is no in-depth analysis of why China should cooperate with BRI countries to cope with climate change. On the other hand, there is little literatures on how China and BRI countries achieve economic win-win, as well as environmental win-win in trade. To fill these gaps, from the perspective of spillover effects, this paper investigates the close economic and carbon emission linkages between China and BRI countries, which is the premise for the two sides to work together to cope with climate change. Based on the above literature analysis, two hypotheses are proposed in this paper.

**Hypothesis** **1.**
*Trade between China and BRI countries brings carbon spillovers to both sides and is significantly heterogeneous between regions and sectors.*


**Hypothesis** **2.**
*Trade between China and BRI countries also brings economic spillover effects to both sides, and is not fully consistent with the aggregate and sectoral characteristics of carbon spillover effects.*


Further, from the perspective of green trade, we put forward policy recommendations for China and BRI countries to jointly cooperate to cope with climate change, and achieve environmental win-win, as well as economic win-win. Accordingly, this study contributes to the literature in two main aspects: First, although it is emphasized that we jointly cope with climate change in the context of the BRI, the links of economic and carbon emissions between China and BRI countries are still unclear. This paper provides empirical evidence on the economic and carbon emissions spillover effects. This contributes to the current literature on the foundations for international cooperation in addressing climate change. Second, due to significant differences in the rate of added value and energy consumption per unit output among the different sectors, the economic spillover benefit and carbon emission spillover costs, from different sectors for the trade partners, may show heterogeneity. This paper investigates the heterogeneity by exploring the differences in economic and carbon emission spillover effects among different sectors. This provides a reference for China and BRI countries to jointly address climate change and achieve economic and environmental win-win, by optimizing trade structure.

## 3. Methodology and Data

### 3.1. Carbon Emissions and Economic Spillover Effects Model

For the tri-regional input–output model, the I-O accounting relationship *x* = *Ax* + *y* becomes as follows:(1)$$\left[\begin{array}{c} X^{C}\\ X^{B}\\ X^{O} \end{array}\right]=\left[\begin{array}{ccc} A^{CC} & A^{CB} & A^{CO}\\ A^{BC} & A^{BB} & A^{BO}\\ A^{OC} & A^{OB} & A^{OO} \end{array}\right]\left[\begin{array}{c} X^{C}\\ X^{B}\\ X^{O} \end{array}\right]+\left[\begin{array}{c} Y^{C}\\ Y^{B}\\ Y^{O} \end{array}\right]$$
where region *C* represents China, region *B* represents BRI countries, and region *O* represents other countries. *X^r^* is the output vector of region *r* (*r* = *C*, *B*, *O*), *Y^r^* is the final demand vector of region *r* (*r* = *C*, *B*, *O*), and *A^rs^* is the intermediate use coefficient matrix of product supply from region *r* (*r* = *C*, *B*, *O*) to region *s* (*s = C*, *B*, *O*). Equation (1) can be morphed into the following equation:(2)$$\left[\begin{array}{c} X^{C}\\ X^{B}\\ X^{O} \end{array}\right]=\left[\begin{array}{ccc} {\left(I-A^{CC}\right)}^{-1} & 0 & 0\\ 0 & {\left(I-A^{BB}\right)}^{-1} & 0\\ 0 & 0 & {\left(I-A^{OO}\right)}^{-1} \end{array}\right]\left[\begin{array}{ccc} 0 & A^{CB} & A^{CO}\\ A^{BC} & 0 & A^{BO}\\ A^{OC} & A^{OB} & 0 \end{array}\right]\left[\begin{array}{c} X^{C}\\ X^{B}\\ X^{O} \end{array}\right]$$

Define *Mrr* = (*I* − *A^rr^*)^−1^ and *D^rs^ = M^rr^A^rs^*, Equation (2) can be written as follows:(3)$$\left[\begin{array}{c} X^{C}\\ X^{B}\\ X^{O} \end{array}\right]=\left[\begin{array}{ccc} 0 & D^{CB} & D^{CO}\\ D^{BC} & 0 & D^{BO}\\ D^{OC} & D^{OB} & 0 \end{array}\right]\left[\begin{array}{c} X^{C}\\ X^{B}\\ X^{O} \end{array}\right]+\left[\begin{array}{ccc} M^{CC} & 0 & 0\\ 0 & M^{BB} & 0\\ 0 & 0 & M^{OO} \end{array}\right]\left[\begin{array}{c} Y^{C}\\ Y^{B}\\ Y^{O} \end{array}\right]$$

Then Equation (3) can be revealed as follows:(4)$$\left[\begin{array}{c} X^{C}\\ X^{B}\\ X^{O} \end{array}\right]=\left[\begin{array}{ccc} F^{CC} & 0 & 0\\ 0 & F^{BB} & 0\\ 0 & 0 & F^{OO} \end{array}\right]\left[\begin{array}{ccc} I & U^{CB} & S^{CO}\\ S^{BC} & I & U^{BO}\\ U^{OC} & S^{OB} & I \end{array}\right]\left[\begin{array}{ccc} M^{CC} & 0 & 0\\ 0 & M^{BB} & 0\\ 0 & 0 & M^{OO} \end{array}\right]\left[\begin{array}{c} Y^{C}\\ Y^{B}\\ Y^{O} \end{array}\right]$$
where *S^CO^* = (*D^CB^D^BO^* + *D^CO^*) (*I* − *D^OB^D^BO^*)^−1^, *U^CB^* = *D^CB^* + *S^CO^D^OB^*, and *F^CC^* = [*I* − *D^CB^ D^BC^* − *S^CO^* (*D^OB^D^BC^* + *D^OC^*)]^−1^. Likewise, the expressions of *F^BB^*, *F^OO^*, *U^BO^*, *U^OC^*, *S^BC^*, and *S^OB^* can be presented.

*M^CC^* can be interpreted as an intra-regional multiplier, representing the impact of a unit of final product in region *C* on its own output through intra-regional industrial linkages. *S^CO^* and *U^CB^* can be interpreted as an inter-regional spillover multiplier, representing the spillover effect of a unit final demand in regions *B* and *O* with output from region *C*. *F^CC^* is an inter-regional feedback multiplier, representing the feedback effect of a unit final demand in region *C* on its own output through cyclical inter-regional economic linkages, with similar explanations for all other variables.

Further, Equation (4) can be decomposed as follows:
(5)$$\left[\begin{array}{c} X^{C}\\ X^{B}\\ X^{O} \end{array}\right]=\left[\begin{array}{c} M^{CC}Y^{C}\\ M^{BB}Y^{B}\\ M^{OO}Y^{O} \end{array}\right]+\left[\begin{array}{c} U^{CB}M^{BB}Y^{B}+S^{CO}M^{OO}Y^{O}\\ S^{BC}M^{CC}Y^{C}+U^{BO}M^{OO}Y^{O}\\ U^{OC}M^{CC}Y^{C}+S^{OB}M^{BB}Y^{B} \end{array}\right]+\left[\begin{array}{c} (F^{CC}-I)(M^{CC}Y^{C}+U^{CB}M^{BB}Y^{B})+S^{CO}M^{OO}Y^{O}\\ (F^{BB}-I)(M^{BB}Y^{B}+S^{BC}M^{CC}Y^{C})+U^{BO}M^{OO}Y^{O}\\ (F^{OO}-I)(M^{OO}Y^{O}+U^{OC}M^{CC}Y^{C})+S^{OB}M^{BB}Y^{B} \end{array}\right]$$

The right side of Equation (5) consists of three parts. The first part represents the intra-regional multiplier effect, the second part represents the inter-regional spillover effect and the third part represents the inter-regional feedback effect.

In general, if the sum of the three effects is considered to be 1, the multiplier effect is greater than 0.90, the spillover effect is greater than 0.09, and the feedback effect is less than 0.01 [46,47,52]. The inter-regional spillover coefficients are far greater than the inter-regional feedback coefficients [58,59]; therefore, we only analyze the spillover effects of carbon emissions between China and BRI countries.

First, the inter-regional carbon emission spillover effects can be measured by adding the carbon emission coefficient vector *E* based on Equation (5). For example, the carbon emission spillover effects (*CSE*) from region *C* to region *B* can be composed as follows:
(6)$$CSE^{CB}=E^{B}S^{BC}M^{CC}Y^{C}$$
where *E^B^* denotes the carbon emission coefficient vectors of *B* regions, which is obtained by dividing the amount of carbon emission by the total output.

Second, the inter-regional economic spillover effects can be measured by adding value-added coefficient vector *E* based on Equation (5). For example, the economic spillover effects (*ESE*) from region *C* to region *B* can be set up as follows:
(7)$$ESE^{CB}=V^{B}S^{BC}M^{CC}Y^{C}$$
where *V^B^* denotes the value-added coefficient vector of the *B* regions, which is obtained by dividing the value-added volume by the total output.

Similarly, carbon emission and economic spillover effects from *B* regions to region *C* are calculated.

### 3.2. Data Sources

Data of multi-regional input–output table and CO_2_ emissions are from the Eora database [60,61,62,63]. Compared to other input–output databases, the Eora database contains the largest number of countries and the most up-to-date data. The latest database includes 190 countries or regions in 2016 (includes ROW area), with each area containing 26 sectors (as shown in Appendix A). In September and October 2013, Chinese President Xi Jinping proposed building the “New Silk Road Economic Belt” and the “21st Century Maritime Silk Road”, respectively, referred to as the Belt and Road Initiative. The Belt and Road Initiative has had an even greater impact in promoting economic cooperation among countries, and has attracted more than 140 countries since its inception [64]. We initially chose the 65 countries that first joined the BRI [65], finally including 61 countries from Asia, Europe, and Africa in the analysis due to the lack of data (removing Timor-Leste, Macedonia, Montenegro, and Palestine), as shown in Appendix B.

## 4. Results and Discussion

### 4.1. Characteristics of Trade between China and BRI Countries

Existing studies have found that trade is closely related to economic growth and carbon emissions. As an important trading partner of BRI countries, bilateral trade cooperation between China and BRI countries has a pulling effect on promoting economic growth along the route. Overall, the total import and export commodities between ALL-BRI and China amounted to US$88.246 billion, of which, the intermediate trade volume was US$66.489 billion and the final trade volume was US$21.757 billion, accounting for 75.34% and 24.66% of the total trade volume between China and BRI countries, respectively.

In terms of trade spatial pattern, AS-BRI has the strongest trade ties with China, followed by EU-BRI, and AF-BRI. In 2016, the total bilateral trade between AS-BRI and China reached US$73.387 billion, accounting for 83.16% of the total trade between BRI countries and China. The total bilateral trade between EU-BRI and AF-BRI and China were US$138.39 billion and US$1.020 billion, accounting for 15.68% and 1.16% of the total trade between BRI countries and China, respectively.

In terms of sectoral structure, China’s bilateral trade with BRI countries shows concentration and symmetry. Specifically, textiles and wearing apparel (S5), petroleum, chemical and non-metallic mineral products (S7), metal products (S8), electrical and machinery (S9), transport (S19), and financial intermediation and business activities (S21), the total trade of these six sectors exceeds 70% of China’s total bilateral trade with BRI countries. Among them, the bilateral trade between China and BRI countries in the electrical and machinery (S9) sector is 25.063 billion USD, and the trade in intermediate goods accounts for 93.94%. The petroleum, chemical and non-metallic mineral products (S7) sector accounted for US$17.586 billion, with 94.16% of the trade in intermediate goods. The metal products (S8) sector accounted for $7.876 billion, with intermediate trade accounting for 67.81%. Other sectors, such as electricity, gas, and water (S13), public administration (S22), and private households (S26) accounted for a lower share of 0.20%, 0.14%, and 0.74%, respectively. The structure of China’s trade with BRI countries by sectors is shown in Figure 1.

### 4.2. Carbon Emissions Spillover Effects

#### 4.2.1. Regional Carbon Emission Spillover Effects

Similar to the above study, we calculated a multiplier effect coefficient of 0.90, a spillover effect coefficient of 0.08, and a feedback effect coefficient of 0.006. We use the method presented in Section 3.1 to measure the mutual CSE between China and BRI countries, the results are presented in Figure 2. First, mutual CSE are evident between China and BRI countries. The CSE from China to all BRI countries is 260.38 Mt, which is slightly lower than those from all BRI countries to China with 275.19 Mt. Second, the largest mutual CSE are between China and Asian BRI countries (AS-BRI), followed by European BRI countries (EU- BRI), and finally African BRI countries (AF-BRI). The CSE from China to AS-BRI, E-BRI, and AF-BRI are 179.21 Mt, 80.17 Mt, and 0.99 Mt, respectively, and the CSE from AS-BRI, EU-BRI, and AF-BRI to China are 228.24 Mt, 42.87 Mt, and 4.07 Mt, respectively. The mutual CSE between China and AS-BRI are the largest due to the large number of AS-BRI countries, and the considerable mutual spillover effect between each AS-BRI and China, on average.

Although CSE between regions are the result of multiple factors, they are primarily related to trade links of intermediate goods (Zhang et al., 2016) First, as demonstrated in Table 1, the volume of intermediate products from China to all BRI countries is US$323.30 billion, which is less than that from BRI countries to China (US$341.59 billion). Therefore, CSE from all BRI countries to China are slightly higher than from China to all BRI countries. Second, whether in terms of total or average or one-way or mutual, the trade volume of intermediate products between China and AS-BRI is the largest, followed by EU-BRI and AF-BRI. For example, the average volume of intermediate goods trade between China and each AS-BRI is US$13.68 billion, exceeding twice the EU-BRI (US$5.49 billion) and AF-BRI (US$5.36 billion) average level. Consequently, bidirectional CSE between China and AS-BRI are the largest, followed by EU-BRI and AF-BRI.

#### 4.2.2. Sectoral Carbon Emission Spillover Effects

Further analysis of the sectoral distribution of mutual CSE between China and BRI countries reveals the following two characteristics, as shown in Figure 3, Figure 4, Figure 5 and Figure 6. First, the sectoral distribution of bidirectional CSE is symmetrical, indicating that the sectors wherein China spills over more carbon emission to BRI countries, are also the sectors wherein BRI countries spillover more carbon emissions to China. Whether between China and all BRI countries or between China and individual BRI countries on various continents, the sector with the highest mutual CSE is electricity, gas, and water (S13), followed by petroleum, chemical, and non-metallic mineral products (S7) (Figure 2). Second, the sectoral distribution of bidirectional CSE is highly concentrated. As shown in Figure 1, CSE from BRI countries to China are concentrated in five sectors, including electricity, gas, and water (S13), petroleum, chemical, and non-metallic mineral products (S7), electrical and machinery (S9), mining and quarrying (S3), and metal products (S8), and vice versa. From the perspective of all BRI countries to China, CSE from these five sectors account for 84.33% of the total. Among them, the proportion of these five sectors in AS-BRI, EU-BRI, and AF-BRI is 85.04%, 76.15%, and 83.23%, respectively, and the proportion of the five sectors from China to all BRI countries is 81.00%. Among them, the proportion to AS-BRI, EU-BRI, and AF-BRI is 80.15%, 82.87%, and 64.40%, respectively.

As noted previously, the magnitude of the spillover effects primarily depends on the closeness of intermediate product trade linkages. Consistent with the concentration and symmetry of CSE, the sectoral distribution of intermediate product trade between China and BRI countries also show strong agglomeration and symmetry. As shown in Table 2, in both China and BRI countries, the five major sectors account for more than 60% of intermediate product trade in all sectors, including mining and quarrying (S3), petroleum, chemical and non-metallic mineral products (S7), metal products (S8), electrical and machinery (S9), and electricity, gas and water (S13). In particular, the three sectors of petroleum, chemical and non-metallic mineral products (S7), metal products (S8), and electrical and machinery (S9) are not only the most intermediate products exported by China to BRI countries, but also the most intermediate products exported by BRI countries to China.

### 4.3. Economic Spillover Effects and Its Comparison with Carbon Emission Spillover Effects

Similar to the carbon emission results, we calculated an economic multiplier effect factor of 0.94, an economic spillover effect factor of 0.054, and an economic feedback effect factor of 0.006. The mutual trade between China and BRI countries not only causes mutual CSE, but also leads to mutual ESE. Analyzing ESE and comparing them with CSE, we find the following characteristics.

First, from the overall perspective, such as carbon emissions, there are mutual ESE between China and BRI countries. Unlike carbon emissions, the ESE from China to all BRI countries are slightly larger than those from all BRI countries to China; the former is US$260.38 billion, and the latter is US$227.25 billion. CSE are not always positively correlated with ESE. Both are affected by multiple factors, such as production technology and carbon emission intensity. The comparison shows that China’s carbon emission coefficient (carbon emission per unit output) is 0.38 kg/US$, which is higher than 0.41 kg/US$ for all BRI countries, and China’s value-added coefficient (value-added per unit output) is 0.40, also lower than 0.48 for all BRI countries. Further comparison of the three continents reveals that the economic spillover effect remains the largest in Asia, followed by EU-BRI, and the smallest is in AF-BRI, which is the same as carbon emissions. The total amount of mutual ESE reaches US$366.58 billion between China and AS-BRI countries, followed by EU-BRI and AF-BRI, with US$116.66 billion and US$4.38 billion, respectively.

Second, from the sectoral perspective, ESE also exhibit the characteristics of concentration, echoing the findings for CSE. Table 3 demonstrates that ESE between BRI countries and China are concentrated in six sectors, including petroleum, chemical and non-metallic mineral products (S7), mining and quarrying (S3), electrical and mechanical (S9), financial intermediation and business activities (S21), metal products (S8), and financial intermediation and business activities (S21). These sectors account for about 70% between all BRI countries, AS-BRI, EU-BRI, AF-BRI, and China.

Third, further comparing the sectoral proportion between CSE and ESE, it can be found that in 26 sectors, the proportion of ESE in 21 sectors is higher than that of CSE, particularly financial intermediation and business activities (S21); and the proportion of CSE in five sectors is higher than that of economic spillover, particularly electricity, gas and water (S13). As shown in Figure 7, the share of mutual economic spillover is 4.23% greater than that of carbon emissions in the financial intermediation and business activities (S21), whereas the share of mutual carbon emissions spillover is 19.68% greater than that of economies in electricity, gas, and water (S13).

Why is the gap of CSE and ESE between above two sectors so different? There is huge difference in the coefficient of carbon emissions (CCE), as well as a discrepancy in the coefficient of value-added (CVA). Table 4 demonstrates that the coefficient of carbon emissions of electricity, gas, and water (S13) is the highest in all sectors in both China and BRI countries, exceeding 3 kg/US$. This is consistent with Wang et al.’s (2021) findings. The coefficient of carbon emissions of financial intermediation and business activities (S21) is far less than electricity, gas, and water (S13) in both China and BRI countries. Furthermore, Table 3 reveals that the coefficient of value-added (CVA) of financial intermediation and business activities (S21) is the highest in China, and ranks the second in AS-BRI and fourth in EU-BRI; however, the coefficient of value-added (CVA) of electricity, gas, and water (S13) is relatively low, ranking medium among all sectors in China and all BRI countries.

## 5. Conclusions

### 5.1. Main Conclusions

The goal of the BRI is to build a community of shared interest, destiny, and responsibility with mutual political trust, economic integration, and cultural inclusion. If all participants involved in the BRI can achieve a win-win situation in economy and environment, it is of great significance to the world’s sustainable development. In order to achieve the above goals, China is working with BRI countries to deal with climate change. Increasingly, trade ties have led to a close relationship between China and BRI countries in terms of economy and carbon emissions. It is of great significance to investigate the mutual carbon emission spillover effects between China and BRI countries for a common future of sustainable development in BRI regions. From the perspective of regions and sectors, we measured and analyzed the two-way carbon emission spillover effects between China and BRI countries, based on an input–output spillover and feedback effects model. The similarities and differences between carbon emission spillover effects and economic spillover effects were further discussed. The following four conclusions were obtained.

First, there are mutual carbon emission spillover effects between China and BRI countries, which is an important basis for them to work together responding to climate change. The carbon emission spillover effects from China to BRI countries are slightly lower than those from BRI countries to China. Two-way carbon emission spillover effects are the largest between China and Asian BRI countries in both total and average, followed by European BRI countries and African BRI countries. The overall characteristics and regional distribution of the above carbon emission spillover effects are closely related to the trade links of intermediate goods.

Second, the sectoral carbon emission spillover effects between China and BRI countries have the characteristics of concentration and symmetry, because of the sectoral agglomeration of intermediate goods trade. The carbon emission spillover effects of both sides are highly concentrated in several sectors, such as electricity, gas, and water (S13), petroleum, chemicals, and non-metallic mineral products (S7), electrical and machinery (S9), mining and quarrying (S3), and metal products (S8). Therefore, China and BRI countries need to focus on these industries in the process of joint efforts to address climate change and reduce emissions.

Third, just like carbon emissions, there are also mutual economic spillover effects between China and BRI countries. The economic spillover effects from China to all BRI countries are slightly larger than those from all BRI countries to China, differing from carbon emissions. The mutual economic spillover effect remains the largest between China and Asian BRI countries, followed by European and African BRI countries. It is of great significance for BRI participants to achieve economic and environmental win-win through trade.

Fourth, as with carbon emissions, economic spillover effects from sectors also exhibit the characteristics of concentration and symmetry, due to the sectoral agglomeration of intermediate goods trade. Notably, the sector with the largest mutual carbon emission spillover effects is not the sector with the largest mutual economic spillover effects. This is primarily related to differences in carbon emission and value-added coefficients between sectors. Therefore, it is necessary to increase trade in sectors with large economic spillover effects, and reduce trade in sectors with large carbon emission spillover effects, in the process of China’s joint efforts with BRI countries to combat climate change.

### 5.2. Policy Implications

It is for this reason that China and BRI countries need to work together to tackle climate change, receiving economic spillover benefits of trade and the costs of carbon emission spillover at the same time. Key to cooperation when dealing with climate change is in how both China and BRI countries can obtain more economic spillover benefits, as well as reducing the cost of carbon emission spillover. China and BRI countries are bound to encounter many obstacles in the process of cooperation in addressing climate change. For example, interests are divided and cooperation is not effectively guaranteed. However, contradiction between economic development and environmental protection is the most prominent. Based on the above reasons, we make the following suggestions.

First, it is suggested that BRI countries strengthen trade links of intermediate products with China, by improving the division of global value chains, which is beneficial to their sustainable development, as they can get more economic spillover effects and assume less carbon emission spillover effects. Moreover, while strengthening trade ties, China and BRI countries can comprehensively strengthen green cooperation, prioritizing the development of renewable energy, and promoting cleaner production technology, which is of great significance to global climate governance. For example, in the future, further strengthening resource cooperation with Laos/Vietnam and other Southeast Asian countries, to develop hydroelectric power generation by making full use of abundant local hydro resources. Also, developing photovoltaic power generation in AS-BRI, where solar energy is abundant.

Second, close trade cooperation has been carried out between China and Asian BRI countries; therefore, trade between China and other BRI countries can be further strengthened in the future. Bilateral trade between China and European BRI countries can focus more on those high-tech and service sectors, with high value-added rates as well as low energy consumption and carbon emissions per unit output. In particular, to further increase cooperation in this area in the future, China should strengthen its cooperation with economically underdeveloped regions, such as Southern Africa. For example, in recent years, Sida Times has started off-grid micro-PV businesses in Kenya and Zambia, and the company is gradually expanding its business to other countries in Africa.

Third, optimizing bilateral trade structure between China and BRI countries is conducive to low-carbon development for both sides. Increased trade between those sectors with high value-added but low-carbon emission coefficients, such as financial intermediation and business activities (S21), is highly recommended. Meanwhile, trade should be reduced between other sectors with a low value-added coefficient but high carbon emissions coefficient, such as electricity, gas, and water (S13). Exploring the possibility of further deepening cooperation, optimizing trade structure can help achieve win-win situations between China and BRI countries; that is, promoting economic development, while also reducing carbon emissions for both sides would be beneficial for both. For example, in March 2022, the Belt and Road International Alliance for Green Development (BRIGAD) conducted an exchange on ASEAN green finance to support green and low-carbon energy transition. 

Fourth, each country should strive to reduce their domestic carbon emission coefficient and increase the value-added coefficient of key sectors, by improving technological and energy efficiency. In the case of the same trade volume, if the country’s domestic carbon emission coefficient is low and the value-added coefficient is high, the country can obtain more economic spillover effects and bear less carbon emission spillover effects than other countries, and this would also have a positive impact on other countries. Of course, the fundamental reason is to improve the level of education in BRI countries, to encourage the innovation of enterprises, and to develop the level of science and technology. Therefore, these are important ways for climate governance of BRI regions—and even the world—that each country improves its technological level and energy efficiency.

### 5.3. Further Study

The authors analyze why and how China and the BRICS countries are working together to address climate change in terms of trade-related carbon emission and economic spillover effects. Further research could be improved in three ways. First, internal coefficients are a key factor affecting the structure of trade [66], and the impact of internal coefficients on spillover effects could be further explored in the future. Second, this study treats final demand as a whole, which may be better if multiple demands, such as household consumption, are treated separately. Third, the relationship between findings and resource decoupling can be further discussed in future research plans.

## Figures and Tables

**Figure 1 ijerph-19-09553-f001:**
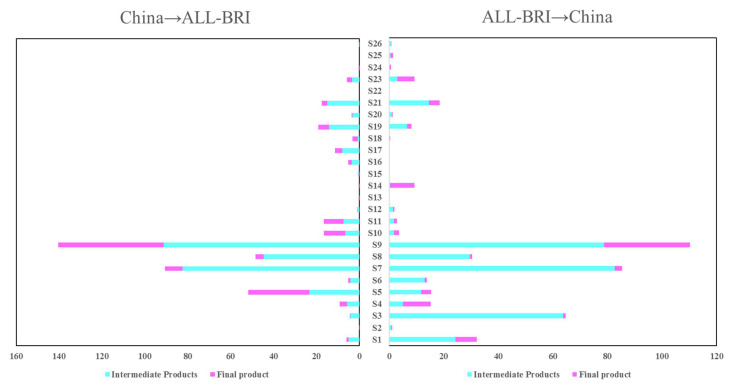
Trade of products between China and BRI countries (US$ billion).

**Figure 2 ijerph-19-09553-f002:**
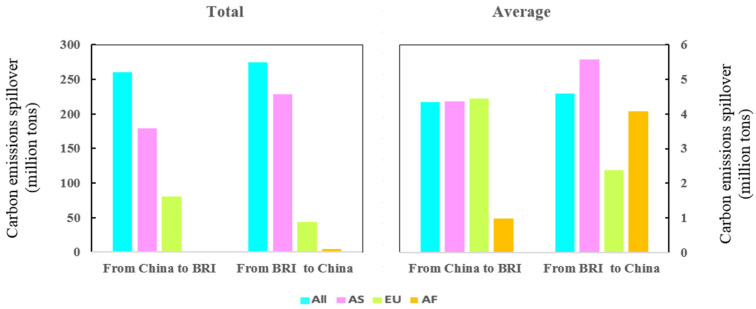
CSE between China and BRI countries.

**Figure 3 ijerph-19-09553-f003:**
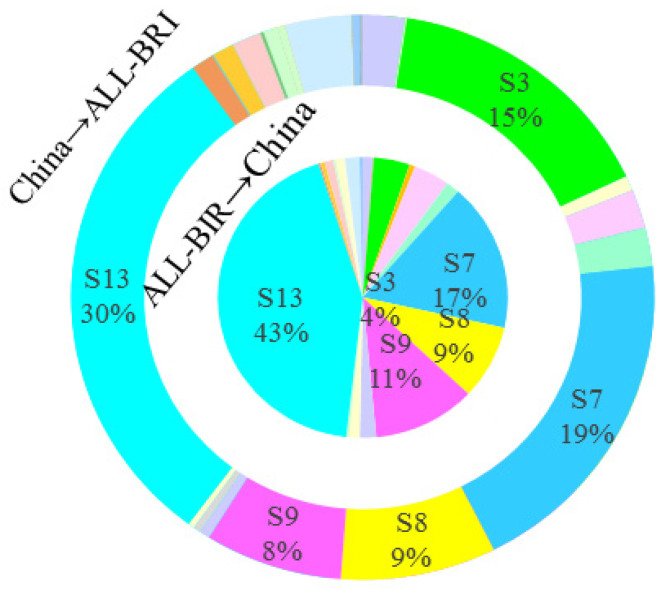
Sectoral CSE between China and ALL-BRI.

**Figure 4 ijerph-19-09553-f004:**
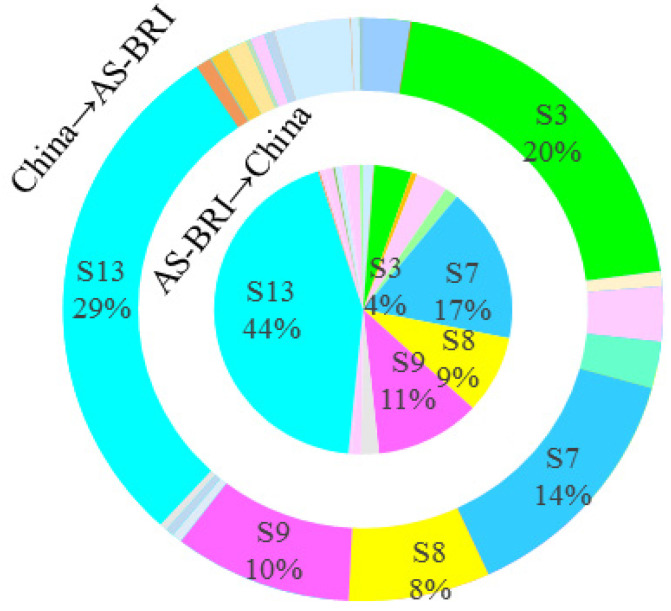
Sectoral CSE between China and AS-BRI.

**Figure 5 ijerph-19-09553-f005:**
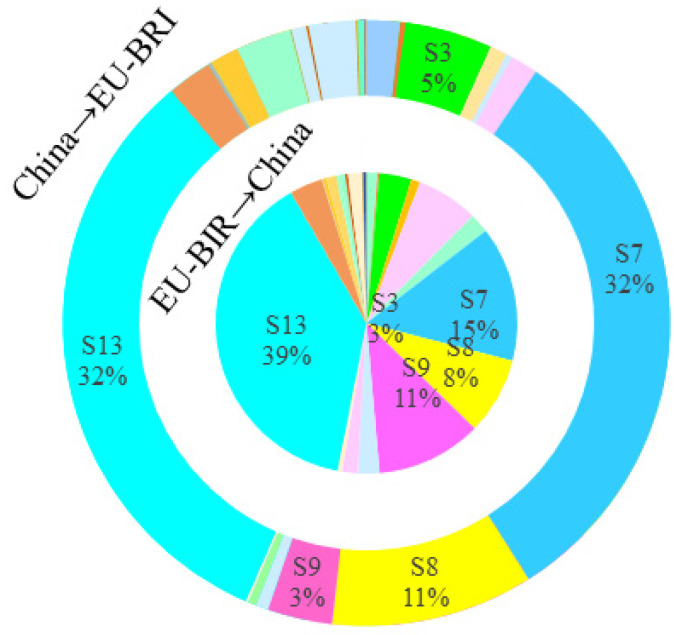
Sectoral CSE between China and EU-BRI.

**Figure 6 ijerph-19-09553-f006:**
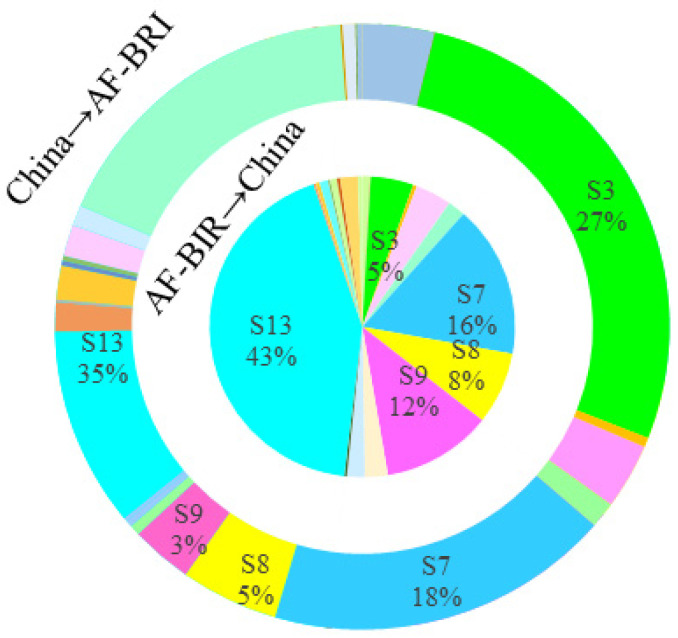
Sectoral CSE between China and AF-BRI.

**Figure 7 ijerph-19-09553-f007:**
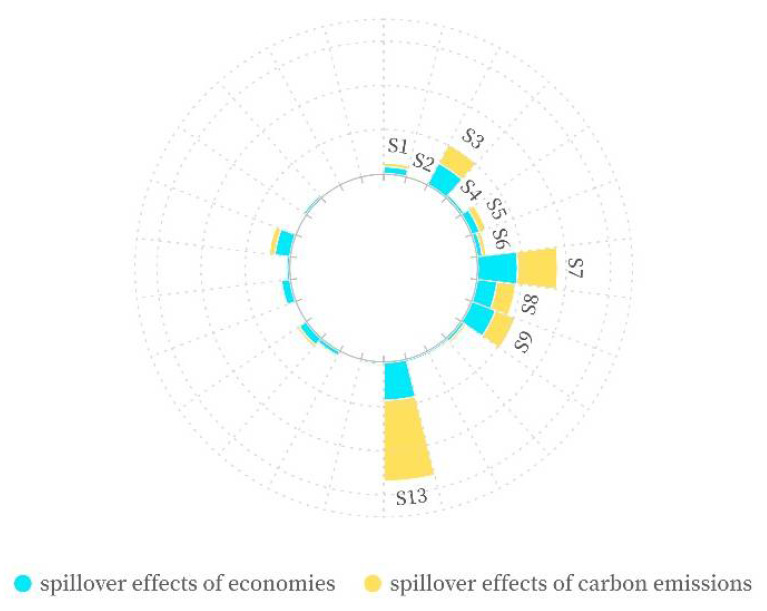
Comparison of CSE and ESE.

**Table 1 ijerph-19-09553-t001:** The intermediate product trade between China and BRI countries (US$ billion).

		BRI→China	China→BRI	China↔BRI
AS	Total	289.63	271.11	560.74
	Average	7.06	6.61	13.68
EU	Total	50.20	48.59	98.79
	Average	2.79	2.70	5.49
AF	Total	1.76	3.59	5.36
	Average	1.76	3.59	5.36
ALL	Total	341.59	323.30	664.89
	Average	5.69	5.39	11.08

**Table 2 ijerph-19-09553-t002:** Sectoral share of intermediate product trade between China and BRI countries (%).

	S3	S7	S8	S9	S13	SO	Sum
China→ALL-BRI	1.30	25.53	13.79	28.19	0.02	31.17	100.00
ALL-BRI→China	2.96	23.83	7.35	21.27	2.77	41.82	100.00
China→AS-BRI	1.28	26.51	13.90	28.28	0.02	30.01	100.00
AS-BRI→China	2.94	24.50	7.04	22.46	2.94	40.12	100.00
China→EU-BRI	1.27	20.14	13.43	28.02	0.03	37.11	100.00
EU-BRI→China	3.00	19.64	9.17	15.04	1.90	51.25	100.00
China→AF-BRI	3.68	24.10	10.75	23.57	0.05	37.85	100.00
AF-BRI→China	5.65	33.95	5.61	3.33	0.73	50.73	100.00

Note: SO present sectors other than S3, S7, S8, S9, and S13.

**Table 3 ijerph-19-09553-t003:** Sectoral share of economic spillovers between China and BRI countries (%).

	S3	S7	S8	S9	S13	S21	SO	Sum
China ↔ ALL-BRI	10.79	17.89	8.91	11.41	17.07	6.85	27.08	100.00
China ↔ AS-BRI	12.72	15.24	8.57	12.75	15.32	7.38	28.02	100.00
China ↔ EU-BRI	4.62	25.94	10.09	7.01	22.91	4.68	24.75	100.00
China ↔ AF-BRI	10.78	16.06	7.61	12.99	4.21	13.33	35.02	100.00

Note: SO present sectors other than S3, S7, S8, S9, S13 and S21.

**Table 4 ijerph-19-09553-t004:** Comparison of CCE and CVA between typical sectors.

	CCE and Its Rank		CVA and Its Rank	
	S13	S21	S13	S21
China	4.55 (1st)	0.27 (10th)	0.21 (19th)	1.44 (1st)
All-BRI	4.80 (1st)	0.27 (10th)	0.44 (16th)	0.69 (2nd)
AS-BRI	5.35 (1st)	0.27 (10th)	0.45 (15th)	0.75 (2nd)
EU-BRI	3.85 (1st)	0.22 (13th)	0.37 (16th)	0.53 (4th)
AF-BRI	3.39 (1st)	0.62 (2nd)	0.85 (2nd)	0.84 (3rd)

Note: CCE presents the coefficient of carbon emissions (kg/US$); CVA presents the coefficient of value-added, and the ranking is based on 26 sectors from high to low.

## Data Availability

Not applicable.

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
