# Peer review of "How Can China and the Belt and Road Initiative Countries Work Together Responding to Climate Change: A Perspective on Carbon Emissions and Economic Spillover Effects"

_ijerph, 2022, doi:10.3390/ijerph19159553_

Round 1

Reviewer 1 Report

I attach the file.

Author Response

Dear Reviewers,
Thanks very much for taking your time to review this manuscript. I really appreciate all your comments and suggestions! Please find my itemized responses in below and my revisions/corrections in the re-submitted files.
Thanks again!

Reviewer 2 Report

The paper describes the economic and ecologic (CO2) overspills of trade partnerships between China and BRI partners. The paper is well written and describes the trade volumes and its impacts.

The paper might be improved by adding the logistic "efforts" of the trade. As far as I understand (this is not clear to me) the numbers describe/include basically the volumes of the trade goods multiplied with the CO2 emissions (my assumption). But the trade logistics are missing. The logistic chain potentially also includes a huge potential for lowering the environmental impact.

Due to the "political" aspect of such a study I recommend to include more international literature ressources to reach a broader acceptance (in case of potential criticism).

Author Response

(The authors gave the same response as above.)

Reviewer 3 Report

Dear Authors, The subject matter is very interesting, but the presented content should be developed:

1. research question "Why should China and the BRI countries join hands to deal with climate change?" seems obvious - if you collaborate with someone, you need to collaborate in all areas relevant to the parties involved;

2. the analyzes carried out do not show how "this study explores how China and the Belt and Road Initiative countries can work together to cope with climate change." - the formulated conclusions are very general and may apply to any broadly understood economic cooperation;

3. there are no research hypotheses formulated on the basis of the literature review and research questions;

4. there is no indication of the weaknesses of cooperation between the parties and the obstacles that could stand in the way of the implementation of the undertaken plans;

5. 4 charts (Figure 2) are illegible - I suggest enlarging them and numbering them separately;

6. editorial error: "this paper analyzes why and how China and the BRI countries ..." - it is not the article that analyzes, but the authors who present the research results in the article;

7. there is no broader presentation of the results of international research;

8. there are no specific recommendations that would be conducive to achieving the set goal - the broadly understood cooperation seems insufficient.

Author Response

(The authors gave the same response as above.)

Reviewer 4 Report

I found the subject interesting and well-presented and I am grateful for the chance to review this paper.

I have one suggestion, which I believe, benefit this paper:

On subsection “4.1.1. Regional carbon emissions spillover effects below line 206 a total of 3 figures and 4 tables are presented without comments or presentation. In fact Figure 2 and Table 2 are mentioned on the next subsection and Table 3 is mention on section 4.2. My suggestion is to reorganize the figures and tables along with the text. In this way it will be easy to read and understand the figures and tables.

Author Response

(The authors gave the same response as above.)

Round 2

Reviewer 3 Report

I am satisfied with the changes made, I accept the article.